# Internal Flow Prediction in Arbitrary Shaped Channel Using Stream-Wise Bidirectional LSTM

Jaekyun Ko †, Wanuk Choi † and Sanghwan Lee *

Department of Mechanical Convergence Engineering, Hanyang University, Seoul 04763, Republic of Korea;
rhworbs1124@hanyang.ac.kr (J.K.); dhksdnr2003@hanyang.ac.kr (W.C.)
* Correspondence: shlee@hanyang.ac.kr
† These authors contributed equally to this work.

**Abstract:** Deep learning (DL) methods have become the trend in predicting feasible solutions in a shorter time compared with traditional computational fluid dynamics (CFD) approaches. Recent studies have stacked numerous convolutional layers to extract high-level feature maps, which are then used for the analysis of various shapes under differing conditions. However, these applications only deal with predicting the flow around the objects located near the center of the domain, whereas most fluid-transport-related phenomena are associated with internal flows, such as pipe flows or air flows inside transportation vehicle engines. Hence, to broaden the scope of the DL approach in CFD, we introduced a stream-wise bidirectional (SB)-LSTM module that generates a better latent space from the internal fluid region by additionally extracting lateral connection features. To evaluate the effectiveness of the proposed method, we compared the results obtained using SB-LSTM to those of the encoder–decoder(ED) model and the U-Net model, as well as with the results when not using it. When SB-LSTM was applied, in the qualitative comparison, it effectively addressed the issue of erratic fluctuations in the predicted field values. Furthermore, in terms of quantitative evaluation, the mean relative error (MRE) for the x-component of velocity, y-component of velocity, and pressure was reduced by at least 2.7%, 4.7%, and 15%, respectively, compared to the absence of the SB-LSTM module. Furthermore, through a comparison of the calculation time, it was found that our approach did not undermine the superiority of the neural network's computational acceleration effect.

**Keywords:** deep learning (DL); computational fluid dynamics (CFD); convolutional neural network (CNN); encoder–decoder (ED); recurrent neural network (RNN); long short-term memory (LSTM)



## 1. Introduction

Engineering advancements in computational fluid dynamics (CFD) have facilitated the comprehension of flow and object–flow interactions. However, even if the pre-processing for numerical simulation, such as mesh generation, the differentiation of governing equations, boundary condition setting, and solver algorithms, is implemented in advance, the high computational cost of calculating the flow and post-processing the results has become an issue. This poses a significant problem for fluid mechanics engineers who need to conduct numerous experiments under diverse conditions.

Hence, neural network (NN) models have emerged as a viable alternative to address the aforementioned issues, particularly in cases where a substantial amount of data has been generated through CFD simulation. NN models [1,2] are trained by adopting the supervised learning strategy, which uses a dataset of paired inputs and the corresponding target obtained through numerous CFD processes. Specifically, various NN structures are designed according to the data type, such as a fully connected (FC) NN for a sampled point set [3], a graph neural network (GNN) for an unstructured grid [4,5], or a convolutional neural network (CNN) [6] for a structured grid [7].

Among these approaches, the CNN-based framework has a well-structured usage environment, demonstrating particular strength in constructing deep NNs [8,9]. Hence,

it is widely used for the interpolation of other types of data in a structured grid [10,11] or when using domain transformation [12]. For instance, Zhang et al. [13] introduced AeroCNN-2, in which the drag and lift coefficients are predicted based on the shape of the airfoil. Viquerat et al. [14] proposed VGG-like networks [15] that can forecast drag forces for arbitrary shapes. Liu et al. [16] presented Shock-Net, which detects shocks in flow and is three times faster than traditional non-deep-learning methods. Deng et al. [17] proposed Vortex-Net, which identifies vortices in shorter execution times with performance similar to that of the instantaneous vorticity deviation (IVD) method [18]. MS et al. [19] introduced a composite CNN-RNN architecture to estimate the viscosity from flow sequences. Liu et al. [20] designed Metric-Net to select a key time step for simulations.

Moreover, accelerating flow generation using DL methods has also become a mainstream research area. In certain instances, researchers have focused on replacing the spatial information processing of the fluid solver with CNNs. For example, Tompson et al. [21] proposed FluidNet, which accelerates the Eulerian fluid simulation by utilizing a CNN to solve a sparse linear system instead of the Euler equation. Xiao et al. [22] presented a CNN-based Poisson solver that employs a hierarchical structure to efficiently represent large sparse matrices. In addition, researchers have examined the combination of recurrent neural networks (RNNs) to learn temporal information from sequential data. Wiewel et al. [23] proposed a long short-term memory (LSTM)-CNN solver to predict the pressure field during the simulation, which is 100 times faster than a regular pressure solver. Hou et al. [24] proposed U-Net-LSTM, which combines U-Net and LSTM to learn the unsteady flow around a submarine, and they showed results with a higher resolution than those of other CNN-LSTM-based data-driven simulators. However, the problem of a model yielding critical errors upon encountering boundary conditions not previously included in the training dataset [25,26] has not been addressed yet.

As a result, research has actively focused on obtaining flow fields through the encoder–decoder (ED) structure to predict steady-state or averaged flows for various shapes and conditions. Guo et al. [1] demonstrated that a fully convolutional NN with an ED model can effectively learn laminar flow data around a car shape inside a rectangular channel. Ribeiro et al. [27] proposed a U-Net model that incorporates skip connections into the ED model to enhance the accuracy in predicting laminar flow results. On the basis of the aforementioned studies, methods have been introduced to predict solutions for Reynolds-averaged Navier–Stokes (RANS) flows around a cylinder [11] or over airfoil shapes [28,29]. Zhou et al. [30] further utilized parameterized boundary information as the input domain through an FC layer. Nonetheless, most of the datasets used by these models are external or the internal flow fields share similarities with the external flow. Notably, these datasets contain an arbitrarily shaped object that is positioned near the center of the channel, with the walls at the top and bottom ends of the input domain remaining parallel.

Therefore, in this study, we created a novel dataset containing various channel shapes and evaluated the training feasibility of the baseline models. Throughout different experiments, it was demonstrated that the baseline models could not properly learn internal flows, such as pipe flows. To solve this problem, we introduced a stream-wise bidirectional (SB)-LSTM module to learn the lateral connections for each pixel. The SB-LSTM module can be easily attached to existing models, thereby effectively increasing the internal flow learning ability. The proposed SB-LSTM module reduced the mean relative error (MRE) of the x-component of velocity, y-component of velocity, and pressure during evaluations by at least 2.7%, 4.7%, and 15%, respectively.

To the best of our knowledge, the proposed approach is the first method to predict a flow in arbitrarily shaped channels with flow separation and corner vortices and ensures high performance. The contributions of this study are summarized as follows.

- An SB-LSTM module is proposed to capture important physical properties of internal flows, whereby model performance is significantly improved by simply connecting the module in the latent space.

- Qualitative and quantitative evaluations conducted on various channel structures demonstrate the excellent performance of the proposed method compared with that of baseline models.

## 2. Methodology

In this section, we first explain the process of dataset generation, illustrate the structure of the proposed SB-LSTM, and then define the loss function used for training.

### 2.1. Arbitrary Channel Shape Generation

In the case of an external flow, there are no major issues in analyzing it by moving the object so that the center of mass is located in the middle of the CFD solving domain. However, in the case of internal flows, where the pipe interacts across the entire domain, the flow pattern changes depending on the order in which they are connected. Since the fully convolutional network produces feature maps only with partial information of the entire image, there can be issues in learning internal flow datasets that have the mentioned characteristics.

We suggest a data generation method to test the learning capabilities. This method allows for the independent generation of the top and bottom surfaces and also allows for the cross-sectional area of the pipe to vary through a distribution function. Additionally, it allows for the presence of steps and bumps at random positions within the domain.

For this, two serial pipes biased up and down from the center were joined together as in Figure 1. Each serial pipe was constructed by connecting a rectangle aligned with the inlet velocity direction and the space between them in a linear manner. We did not use Bezier curves or sine functions when designing the shape because the flow flowed parallel to the wall without flow separation or vortex generation along this function shape. Examples created through this method are shown in Figure 2.

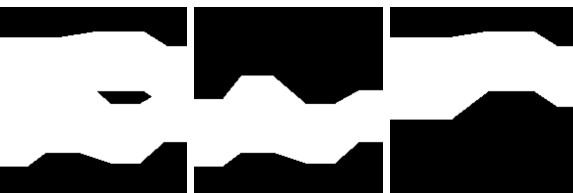

**Figure 1.** Example of upper and lower serial pipe in dataset.

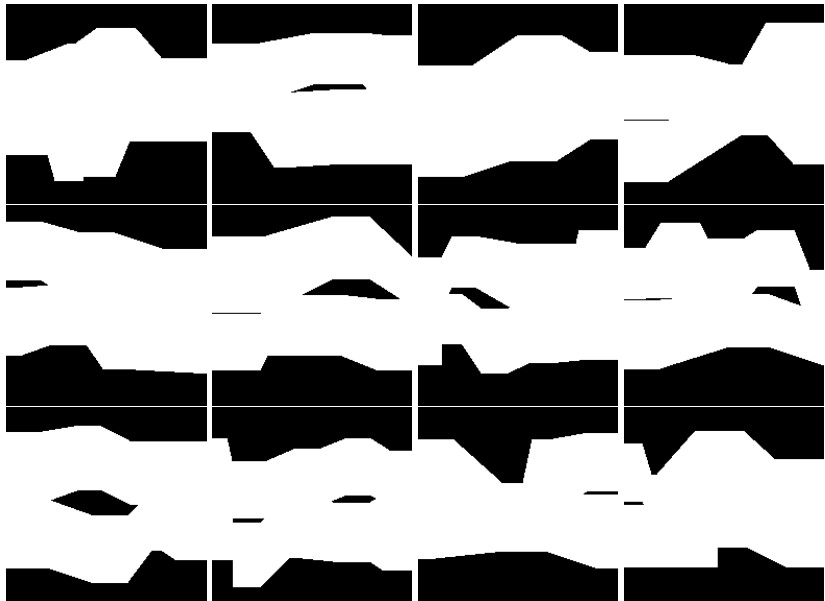

**Figure 2.** Samples of generated channel.

By configuring the pipe as above, we created different flow rates over converging and diverging channels, bumps and cavities at the walls, and objects in the channels. The height *H*, width *W*, and the *x* and *y* positions of each pipe were chosen as the independent variables to be determined before constructing a biased serial pipe, using discrete uniform and normal distributions. Detailed specifications are shown in Figures 3 and 4.

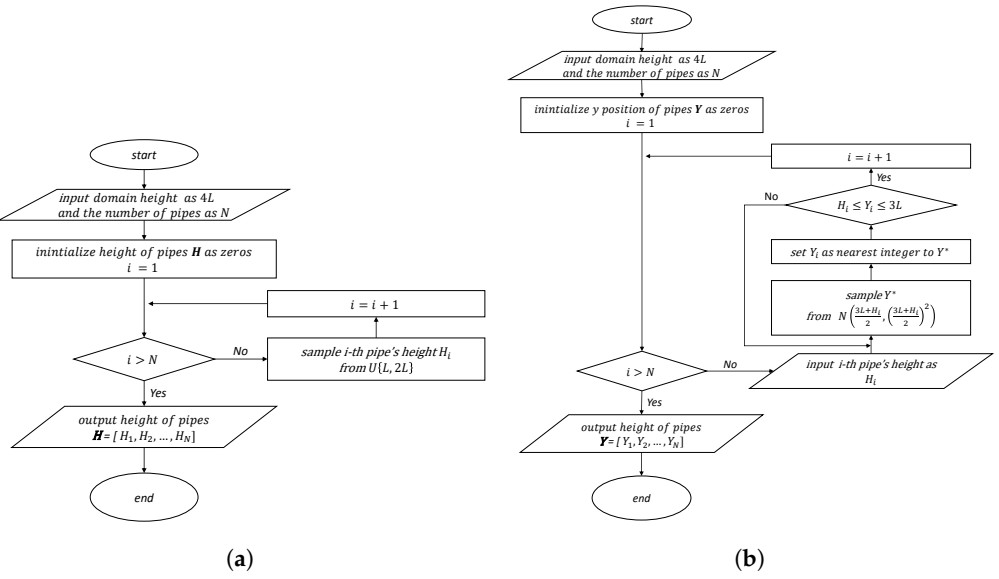

(**a**)          (**b**)

**Figure 3.** Flow diagram for determination of the height and y positions of rectangles consisting of lower serial pipes. From left to right: (**a**) height, (**b**) y position. The superscript * represents a variable used as an intermediate step to determine the actual value.

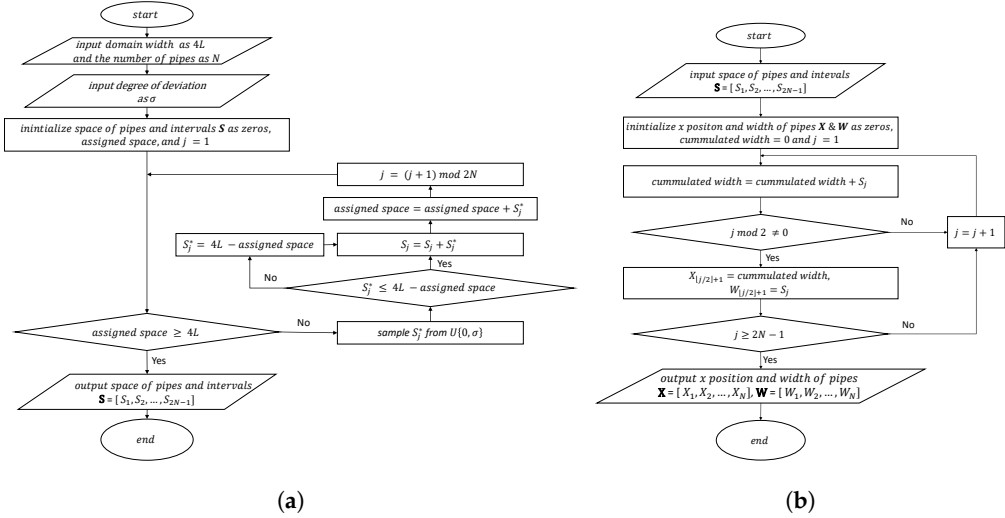

(**a**)          (**b**)

**Figure 4.** Flow diagram for determination of the Width and x positions of rectangles consisting of lower serial pipes. From left to right: (**a**) spaces of pipes and intervals, (**b**) x position and width of pipes. The superscript * represents a variable used as an intermediate step to determine the actual value.

The size of the domain used in the experiment was set to $192 \times 192$. The number of pipes was selected by generating a random integer between three and seven, and the hyperparameter $\sigma$ was set to 20. In addition, to create a higher serial pipe, the lower serial pipe was inverted in the *y*-direction, following the aforementioned procedure.

### 2.2. Target Field Data Generation

An open-source CFD tool, OpenFOAM [31], was employed to generate a target dataset. For mesh generation, a domain with a channel shape of $192 \times 192$ was converted into the 3D 192 mm $\times$ 192 mm $\times$ 1 mm domain of the voxel file that was obtained through the method explained in Section 2.1. The voxel file was finally converted into an STL file and the mesh was created by cutting out the basic domain generated by blockMesh to match the shape of the channel using an STL file, without the refinement process.

In our problem setup, when gravity is neglected, a fluid with kinematic viscosity $\nu$ of $1 \times 10^{-4}$ m$^2$/s enters the channel from the left side as a uniform flow with an x-component velocity $u_{\text{inlet}}$ of $1 \times 10^{-3}$ m/s. The fluid then flows through the 2D channel, where the velocity at the boundaries and the pressure gradient in the normal direction are zero. Finally, the fluid exits through the right side of the channel, where the gauge pressure $p$ is equal to 0 *Pa*. Based on these boundary conditions, the continuity and momentum equations are expressed as follows:

$$\nabla \cdot V = 0 \tag{1}$$

$$(V \cdot \nabla)V = -\nabla P + \nu \nabla^2 V \tag{2}$$

$$P = \frac{p}{\rho_0}, \tag{3}$$

where $\nabla$, $V$, $P$, and $\rho_0$ denote the *del* operator, velocity vector, kinematic pressure, and constant mass density of the fluid, respectively. Our models were set to the laminar because our inlet velocity was low enough. To generate the target data, the SIMPLE algorithm [32] was used for the CFD solution. During post-processing, the corner vortex region near the wall was calculated as the region of interest (RoI) to compare the accuracy at the region. A sample set of fields in the target data is shown in Figure 5, where the channel region is highlighted in black to enhance the distinction between the wall and fluid regions.

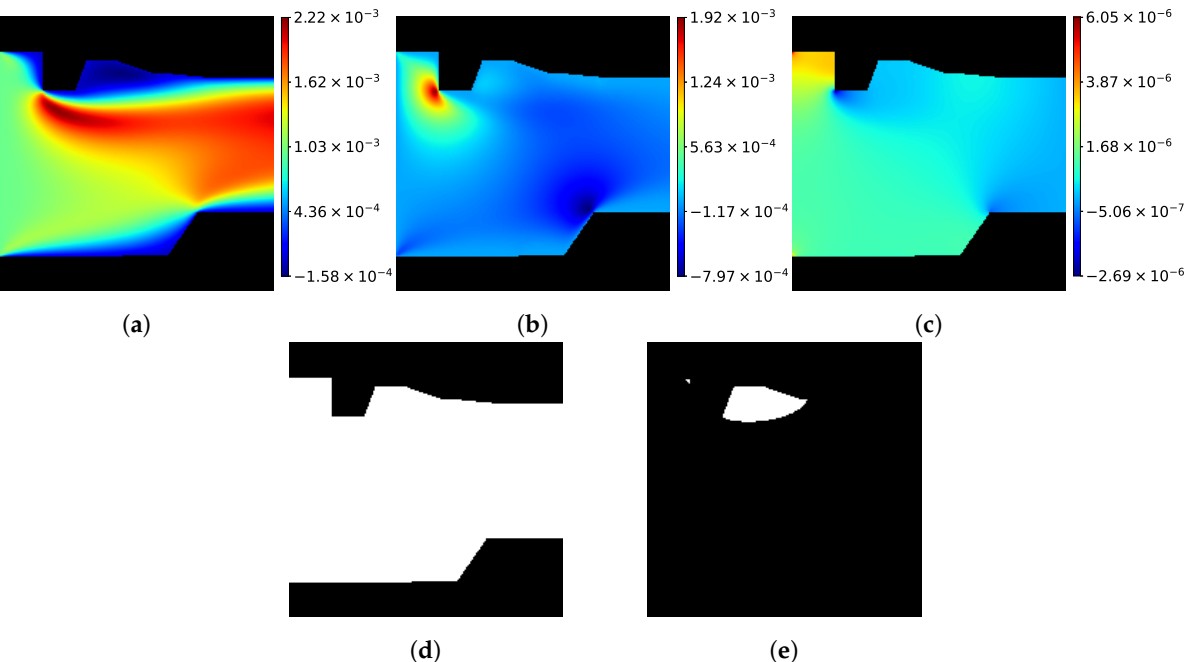

**Figure 5.** Sample set of target field data. From left to right: (**a**) x-component velocity, (**b**) y-component velocity, (**c**) kinematic pressure, (**d**) all fluid region, (**e**) corner vortex region.

### 2.3. Stream-Wise Bidirectional LSTM

The deformation of objects in previous internal flow prediction studies only has a small impact on the distribution of field values, and the overall pattern of the field does

not change significantly. However, the data generated through the above method undergo significant changes depending on the shape of the entire domain. Therefore, if a fully convolutional neural network is used for inference and training, problems arise.

To solve the problems of the existing network model, a module was designed based on bidirectional LSTM, which has shown strength in filling in words through context in natural language processing [33]. Generally, when dealing with dynamic data in deep learning, LSTM has been mainly used for time series data learning [34]. However, there have been studies using the concept of bidirectional LSTM for character inference within images in the field of image processing [35,36] The SB-LSTM module was inspired by this CNN-RNN combined approach.

The SB-LSTM module was designed to take the compressed latent geometry space as input and produce a feature map that includes the relationships between the forward and backward sequences of the feature space. While an encoder with convolutional layers can contract the input and extract high-level features, it does not include information about the lateral connections between pixels [37]. This means that global features related to the shape and flow direction of the channels are not well preserved in latent space when using only an encoder. To address this, the authors propose using bidirectional LSTM at the latent space to learn the directional relationships between pixels.

The entire process of the SB-LSTM module is depicted in Figure 6. The latent geometry space of size $H \times W \times C$, denoting the height, width, and channel, respectively, is first rearranged in the stream-wise direction (along the width axis) through a flattening process. This results in $W$ divided components, each represented as a 1D matrix of size $C \times H$. Second, each serial vector $Z_i$ continuously updates the cell state and hidden state of the forward LSTM, which are initialized to zero with a size equal to $C \times H$, and returns an output $Z'_{f,i}$, where $i$ denotes the index of each input vector. Consequently, this allows us to obtain the output $Z'_{f,i}$, which contains information on the feature spaces sequentially passing along the flow direction to the $i$-th point. Similarly, $Z_i$ is used to update the cell and hidden states of the backward LSTM from the outlet to the inlet direction, outputting $Z'_{b,i}$. Third, the forward and backward hidden states at the $i$-th latent position are combined to create a set of new serial vectors containing the expected lateral relation along all separated areas. Since vectors from the forward and backward LSTM are aggregated together, the length of the output vector becomes $2C \times H$. Finally, the output is repeated $H$ times in the vertical direction in order to share the same information among the latent spaces that are involved in the flattening process. In summary, the proposed SB-LSTM module addresses the local limitations of the latent space in the existing encoder by generating global features through bidirectional LSTM. This makes it suitable for the analysis of internal flows on arbitrary channel shapes. The effect of the module is thoroughly analyzed in Section 3.

### 2.4. Network with Stream-Wise Bidirectional LSTM

The integration of the SB-LSTM module in the network is explained using the example of a module with SB-LSTM applied to U-Net [27], used later in the experiments. The structural overview of U-Net with SB-LSTM is shown in Figure 7. In the middle part of the structure, where encoding ends, the latent space not only serves as an input for X2Nearest Neighbor interpolation, indicated by the blue arrow, for the first decoding process, but also acts as an input for SB-LSTM to extract additional lateral connection information. To adjust the size for application in the network, $1 \times 1$ convolution and nearest neighbor interpolation have been added to the path for SB-LSTM. The added lateral information is concatenated with the latent space and used as an input for the convolution operation in the decoding process, along with the latent space of the original path. The detailed process of the operation of this network is described below.

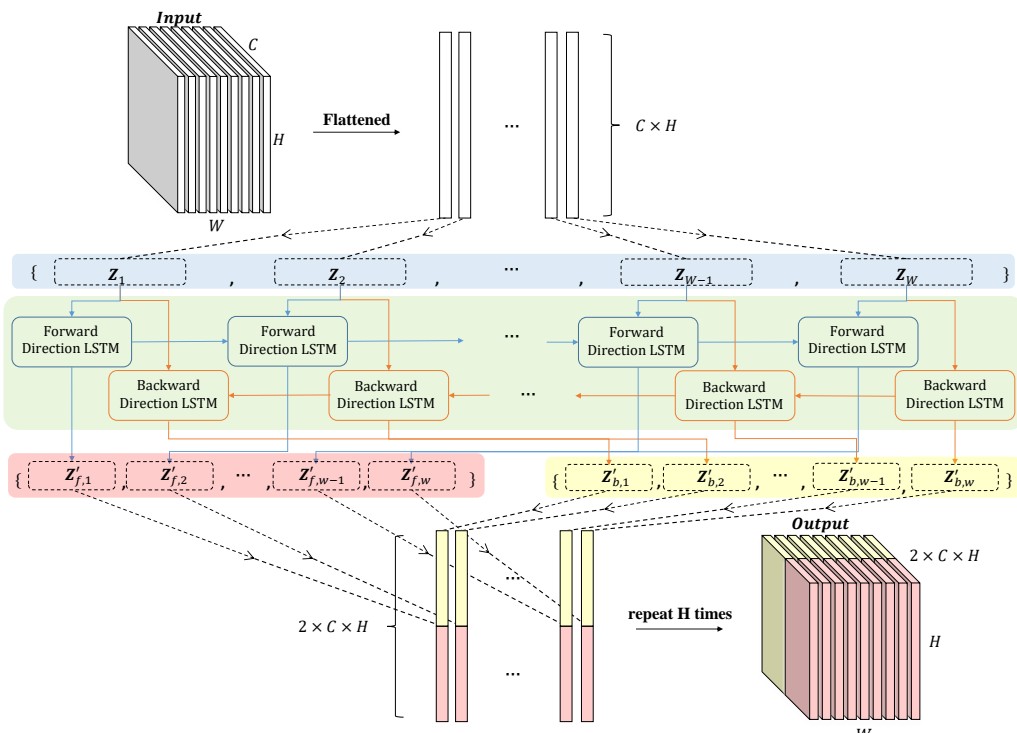

**Figure 6.** Structural overview of the proposed SB-LSTM module.

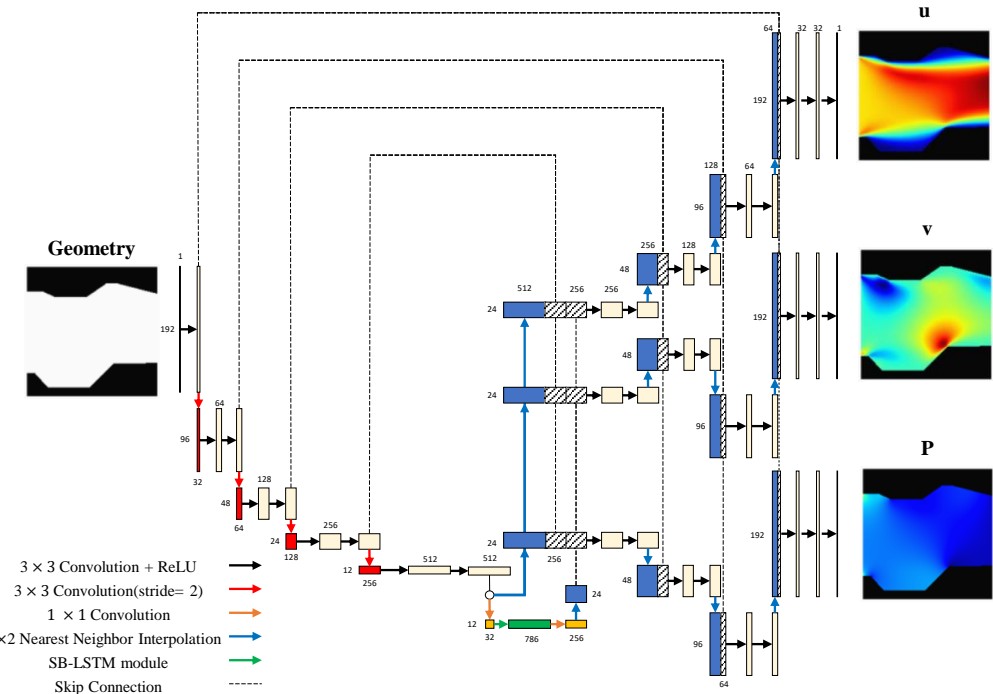

**Figure 7.** Structural overview of the proposed U-Net with SB-LSTM.

Given an input channel shape with dimensions of $192 \times 192 \times 1$, denoting the height, width, and channel, respectively, the model first expands the number of channels to 32 using a $3 \times 3$ convolutional layer with a rectified linear unit (ReLU) activation function. Then, a $3 \times 3$ convolutional layer with a stride of two downsamples the height and width by half and increases the number of channels to 64 using two $3 \times 3$ convolutional layers with ReLU, which yields a $96 \times 96 \times 64$ sized feature map. The same process is applied to the feature map until it reaches the size of $12 \times 12 \times 512$. At the latent space, the number of

channels is reduced to 32 using a $1 \times 1$ convolutional layer and the compressed information is fed into the SB-LSTM module. After receiving the geometrical connection features from the SB-LSTM module, the channels are decreased to 256 using a $1 \times 1$ convolutional layer. The height and width are then upsampled using nearest neighbor interpolation with a scale factor of two. We have selected nearest neighbor interpolation to utilize the relations of adjacent pixels for a continuous upsampling process. The feature map shortly before the SB-LSTM module is also upsampled and both outputs are concatenated channel-wise. Moreover, the feature map from the encoder with the same spatial size is also aggregated to provide channel shape information. Two $3 \times 3$ convolutional layers with ReLU are then sequentially used for feature extraction and channel size reduction. This mechanism is applied repeatedly until the shape of the final output becomes $192 \times 192 \times 32$. Finally, a $3 \times 3$ convolutional layer is used to adjust the number of channels from 32 to 1. To predict $u$, $v$, and $P$, three decoders with the same architecture are employed, all sharing the features extracted by the SB-LSTM module.

### 2.5. Loss Function

Since the velocity $x$-component $u$, velocity $y$-component $v$, and pressure $P$ consist of continuous values, the prediction of each field is similar to a regression task. Hence, the proposed model is trained by minimizing the difference between the predicted and target field at each pixel and the loss function is formulated as

$$Loss = \frac{1}{HW} \sum_{i}^{H} \sum_{j}^{W} \left( \|u_{pred_{i,j}} - u_{target_{i,j}}\|_1^1 + \|v_{pred_{i,j}} - v_{target_{i,j}}\|_1^1 + \|P_{pred_{i,j}} - P_{target_{i,j}}\|_1^1 \right), \quad (4)$$

where $\| \cdot \|_1^1$ represents the mean absolute error (MAE) and subscripts $i$ and $j$ denote the positions of the row and column, respectively.

## 3. Experiments

### 3.1. Dataset and Pre-Processing

A total of 5000 channel shapes were created through the process described in Section 2.1. The 5000 samples were then flipped upside down to double the quantity to 10,000. For all shapes, the samples were used as input into the NN, where pixels inside the binary array indicated the wall and fluid regions by 0 and 1, respectively. The data values obtained using the CFD tool in the domain were converted into target data by assigning the simulation results at the center position of each pixel set in the fluid region to an empty array with dimensions of $192 \times 192$. As a result, we constructed a dataset with 10,000 sets of an input channel shape and corresponding target fields. For the training, validation, and test datasets, we randomly selected and divided the generated samples into 8000, 1000, and 1000 cases, respectively.

During pre-processing, channel normalization was applied to the target field. Each pixel was divided by the standard deviation obtained from corresponding fluid regions to generate non-dimensional values. The pre-processing task is expressed as follows:

$$u^* = \frac{u}{u_{std}}, \quad v^* = \frac{v}{v_{std}}, \quad P^* = \frac{P}{P_{std}}. \quad (5)$$

### 3.2. Comparison Methods

The effectiveness of the SB-LSTM module was evaluated on two baseline models, ED and U-Net. These two models were adopted because they were used to train internal flow datasets in [1] and [27] and have also been referenced in many other papers regarding obtaining steady state solutions. ED consists of an encoder and decoder, and U-Net is an extension of ED with skip connections. In both models, the SB-LSTM module is attached at the latent space, and the structural comparison of the four different methods is depicted in Figure 8.

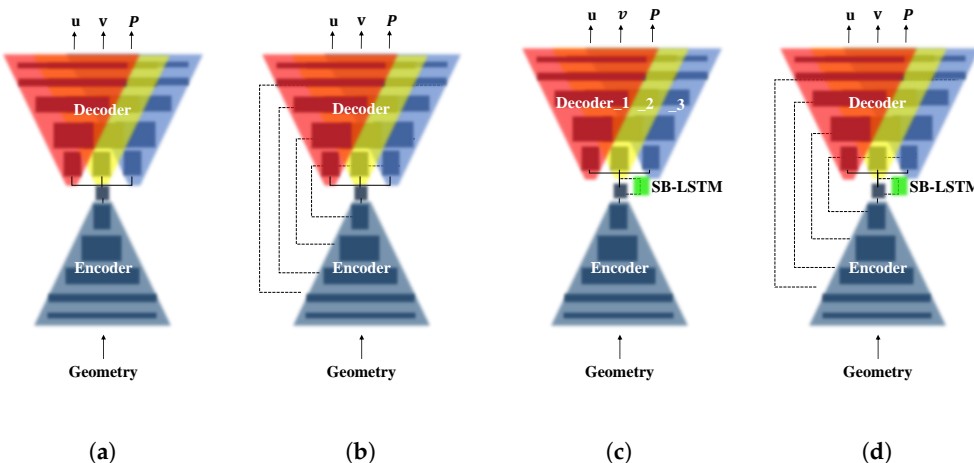

**Figure 8.** Structural comparison of four different methods. From left to right: (**a**) ED, (**b**) U-Net, (**c**) ED with SB-LSTM, (**d**) U-Net with SB-LSTM.

Moreover, the number of model parameters of the four different methods is listed in Table 1. In both ED and U-Net, the model capacity is increased to 4,348,192 parameters with relative increment ratios of 34.67% and 22.60%, respectively. The significant growth in parameters occurs due to the use of the fully connected operation in the input gate, output gate, and forget gate of LSTM. Furthermore, since the network that we use utilizes two LSTM models, one for backward and one for forward processing, the increase in the number of parameters mentioned earlier is doubled. Despite the larger model capacity, overfitting did not occur during training and the models with the SB-LSTM module showed better performance, which is comprehensively analyzed in Sections 4.1–4.3.

**Table 1.** Number of model parameters for the four different methods.

| Method | Number of Model Parameters |
|:---:|:---:|
| ED | 12,540,227 |
| U-Net | 14,890,307 |
| ED with SB-LSTM | 16,888,419 |
| U-Net with SB-LSTM | 19,238,499 |

### 3.3. Implementation Details

In the experiments, we used the Adam optimizer [38] ($\beta_1 = 0.5$, $\beta_2 = 0.999$). The learning rate was initialized to $1 \times 10^{-4}$ and the batch size was set to 64. All methods were trained for 200 epochs and evaluated at each epoch. Using the cosine annealing algorithm [39] as the learning rate scheduler, the learning rate gradually decayed to $1 \times 10^{-6}$. The PyTorch framework [40] and two NVIDIA RTX 3090 GPUs with 24 GB of RAM, 48 GB in total, were used for the implementation. The official code can be found in the following link: https://github.com/choiwanuk/SB-LSTM, accessed on October 19 2023.

## 4. Results and Discussion

### 4.1. Training History

To demonstrate whether the learning progressed well under the above conditions, we plotted four graphs that contained the history of the training and validation loss of the comparison methods. As illustrated in Figure 9, the baseline models, i.e., ED and U-Net, converged well, eventually showing lower validation loss compared to training loss. For methods with the SB-LSTM module, they also exhibited well-trained curve features where no overfitting problems occurred despite the model parameter increase. During the training process, the model with SB-LSTM attached showed a faster decrease in loss per epoch.

While the baseline methods did not show a decrease in loss from around 125 epochs, the attached models maintained a steady decrease.

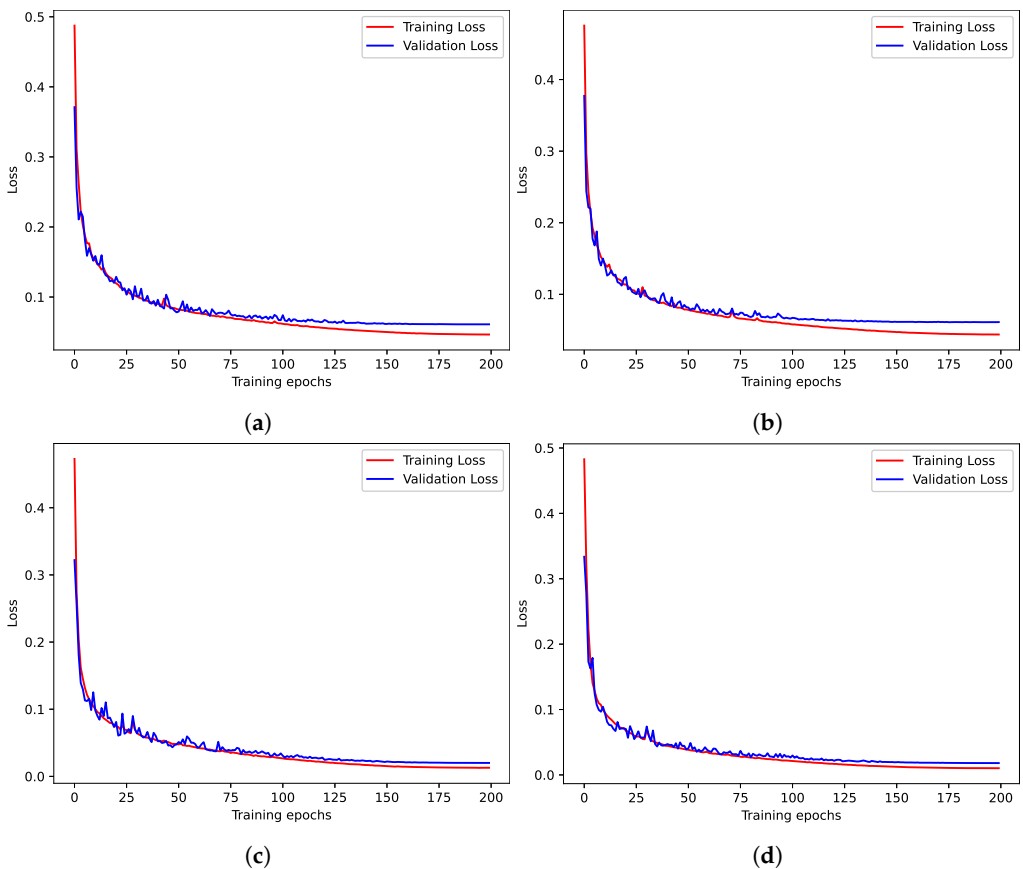

**Figure 9.** Training history of four different methods. From top left to bottom right: (**a**) ED, (**b**) U-Net, (**c**) ED with SB-LSTM, (**d**) U-Net with SB-LSTM.

*4.2. Qualitative Evaluations*

The target field predictions of the four different models are visualized in Figures 10–14. The representative data in each figure are for cases with a nearly constant vertical cross-sectional area, converging, diverging, converging–diverging, and diverging–converging, respectively. The colorbar range was determined by calculating the minimum and maximum values of predictions and targets for each case. In this section, the focus is mainly on the differences between the models with and without the SB-LSTM module, as the differences between the models that include the SB-LSTM module are not clearly visible within the range that encompasses the results of all four models. The qualitative comparison between models with SB-LSTM is visualized in Figures A1–A5 of Appendix A.

As shown in Figure 10, both the ED and U-Net models encountered a problem with a sudden discontinuity appearing in the $u$ field at approximately three quarters of the width from the inlet in the x-direction. This indicates that the models struggled to accurately estimate the flow rate near the outlet of the channel shape. The issue with the $v$ field was not as severe in this interval, instead appearing as a blurry error in the form of a thin line. As for the $P$ field, as it approached the inlet from the outlet, the predicted value deviated more and there was a sudden change in value at around one quarter of the width from the inlet in the x-direction. Additionally, in Figures 11 and 12, it can be observed that the error in the predicted value increases as $u$ approaches the outlet and $P$ approaches the inlet.

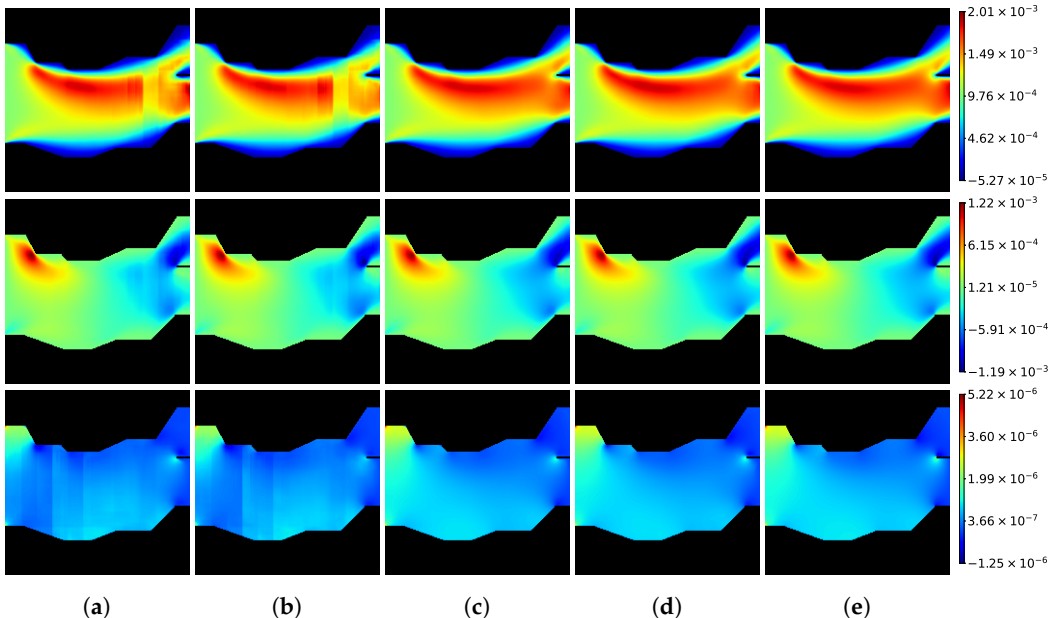

**Figure 10.** Qualitative results of four different methods. From top to bottom: $u$, $v$, $P$. From left to right: (**a**) ED, (**b**) U-Net, (**c**) ED with SB-LSTM, (**d**) U-Net with SB-LSTM, (**e**) target.

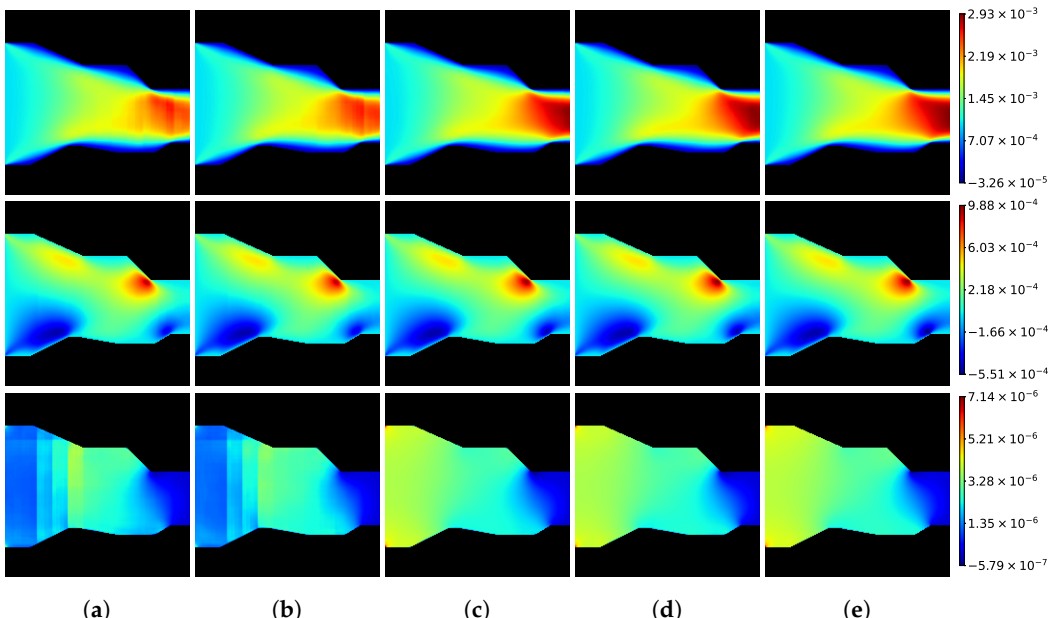

**Figure 11.** Qualitative results of four different methods. From top to bottom: $u$, $v$, $P$. From left to right: (**a**) ED, (**b**) U-Net, (**c**) ED with SB-LSTM, (**d**) U-Net with SB-LSTM, (**e**) target.

These limitations are observed in the convolutional neural network (CNN) components of ED and U-Net. The structure of the CNN is inspired by the visual cortex in animals [41,42] and it thus has the ability to learn and generate high-level feature maps. However, a drawback of the fully convolutional network is that it only considers information within the receptive field range. This means that if the necessary information for the prediction of the target value is outside of this range, the learning process may be flawed and solutions may be generated from irrelevant features. Consequently, when $u$ and $P$ move away from the fixed boundary condition, the results may show tendencies that are not present in the training dataset, as seen in Figure 14, where $u$ suddenly increases despite there being no possibility for a further speed increase.

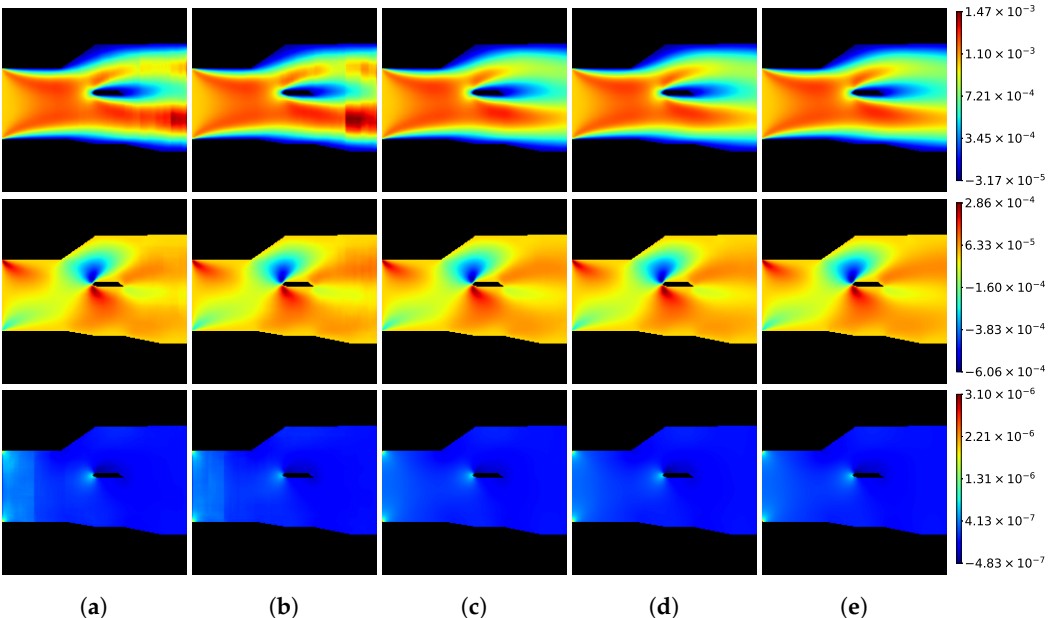

**Figure 12.** Qualitative results of four different methods. From top to bottom: $u, v, P$. From left to right: (**a**) ED, (**b**) U-Net, (**c**) ED with SB-LSTM, (**d**) U-Net with SB-LSTM, (**e**) target.

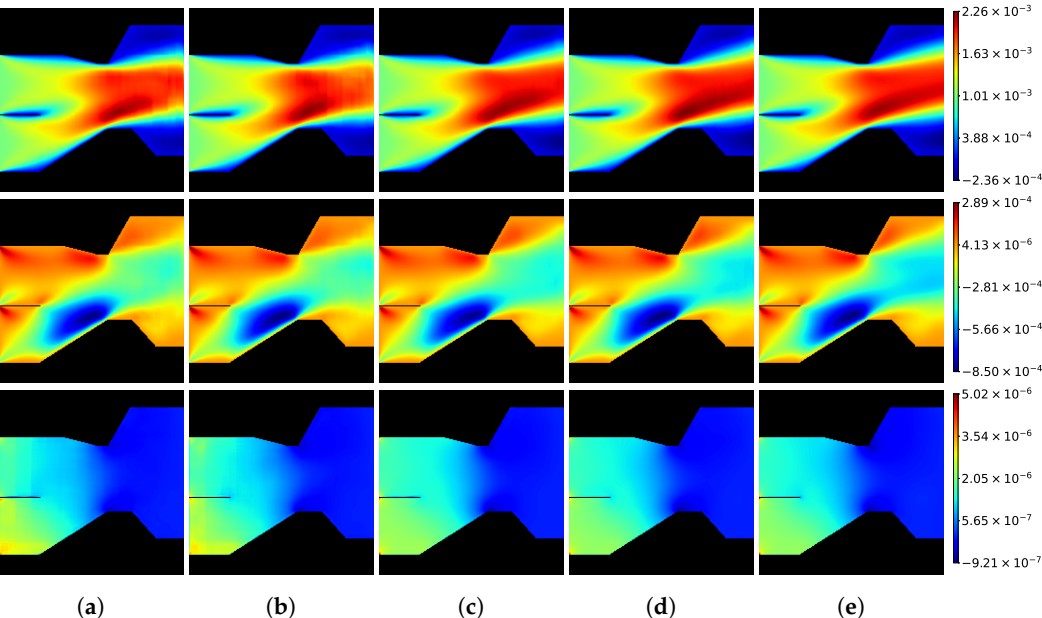

**Figure 13.** Qualitative results of four different methods. From top to bottom: $u, v, P$. From left to right: (**a**) ED, (**b**) U-Net, (**c**) ED with SB-LSTM, (**d**) U-Net with SB-LSTM, (**e**) target.

The proposed network, when combined with the SB-LSTM module, produced convincing results without encountering these types of issues in all the cases that were included in our test dataset. This confirms that not only errors related to patterns that deviate from fluid physics were successfully addressed, as shown in Figures 10–12 and 14, but also the resolution of the rows in Figure 13 was improved. This result can only be achieved by providing and receiving the appropriate features for estimation through lateral connections during the forward process using the attached SB-LSTM module. In other words, this demonstrates that the neural network that we designed possesses a structure that effectively learns the information to transfer during the backpropagation process. In addition to the above results, a qualitative evaluation related to the direction of LSTM in the SB-LSTM

module is described in Appendix B, along with Figure A6 on the training histories and Figure A7 on the results of the network depending on the direction setting of LSTM.

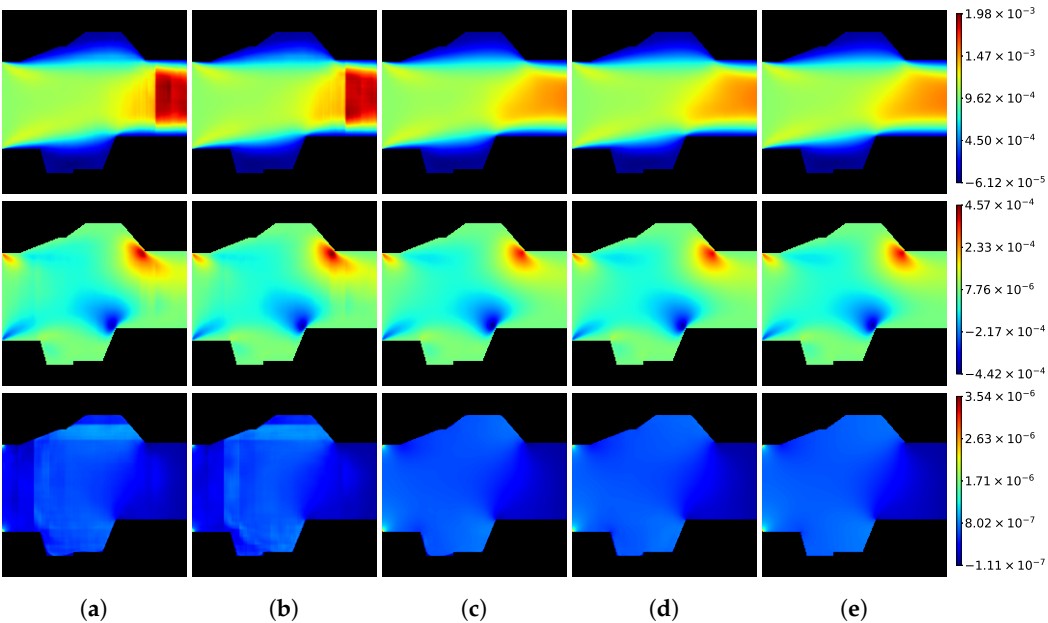

**Figure 14.** Qualitative results of four different methods. From top to bottom: *u*, *v*, *P*. From left to right: (**a**) ED, (**b**) U-Net, (**c**) ED with SB-LSTM, (**d**) U-Net with SB-LSTM, (**e**) target.

*4.3. Quantitative Evaluations*

4.3.1. Mean Relative Error

To quantify and compare the performance of the models, the mean relative error between the prediction and ground truth was calculated for two areas, the entire channel region and the corner vortex region generated near the wall, using the formula in Equation (6):

$$MRE(\alpha, \delta) = \frac{1}{N} \sum_{l=1}^{N} \frac{\sum_{i=1}^{Nx} \sum_{j=1}^{Ny} |\alpha_{ij} - \hat{\alpha}_{ij}| \delta_{ij}}{\sum_{i=1}^{Nx} \sum_{j=1}^{Ny} |\alpha_{ij}| \delta_{ij}}. \tag{6}$$

where $\alpha$, $\delta$, $N$, $i$, $j$, and ˆ denote a scalar field variable, binary field values of the RoI, the number of test datasets, the index values of the width and height axes, and the predicted value, respectively. Table 2 summarizes the mean relative error for each RoI of each model. In the table, the best performance results are shown in bold, and the second-best performance results are underlined for emphasis. U-Net with SB-LSTM showed the best performance for all metrics, followed by ED with SB-LSTM, which showed the second-best performance for all metrics. Note that there was a significant improvement in the case of *P* prediction compared to the decrease in the error rate of the *u* and *v* values. This demonstrates the effective performance improvement of internal flow learning with the SB-LSTM module. These findings align with previous studies that have shown that the U-Net structure outperforms the ED structure in original image processing and flow prediction.

**Table 2.** Quantitative comparisons of mean relative error between four different methods, with ↓ indicating that lower is better. The best result is highlighted in **bold** and the second-best result is underlined.

| Method | MRE (All) (↓) | | | MRE (Corner) (↓) | | |
|---|---|---|---|---|---|---|
| | *u* | *v* | *P* | *u* | *v* | *P* |
| ED | 4.099% | 10.200% | 20.524% | 54.007% | 48.999% | 35.241% |
| U-Net | 4.056% | 10.102% | 20.962% | 53.675% | 48.794% | 35.183% |
| ED with SB-LSTM | <u>1.370%</u> | <u>5.591%</u> | <u>5.019%</u> | <u>30.738%</u> | <u>31.814%</u> | <u>15.831%</u> |
| U-Net with SB-LSTM | **1.195%** | **5.212%** | **4.524%** | **30.350%** | **29.102%** | **13.724%** |

4.3.2. Mean Absolute Error along Height Axis in Entire Channel

To analyze the characteristics of the errors of the models in Section 4.2, the mean absolute error along the height axis was calculated for each width position for the entire test data set using Equation (7).

$$MAE(\alpha, i) = \frac{1}{N} \sum_{l=1}^{N} \sum_{j=1}^{Ny} |\alpha_{ij} - \hat{\alpha}_{ij}| \delta_{ij}. \tag{7}$$

The calculated results are visualized in Figure 15. In the graph, U-Net with SB-LSTM shows the lowest error at all width positions, followed by ED with SB-LSTM, which has the next lowest error. Regardless of the presence or absence of the SB-LSTM module, the error of all models tends to increase as the predicted regions move further away from the boundary, with the boundary condition set to a fixed value. More attention should be given to models trained without the SB-LSTM module because of the periodic and significant increase in error in the stream-wise case. These peaks occur with exactly the same spacing and position, regardless of whether they are close to or far from the fixed boundary conditions in all width ranges of $u$, $v$, and $P$. This indicates that the error observed in Section 4.2 is occurring throughout the entire area, suggesting a problem with the approach of learning using a fully connected NN. However, when the SB-LSTM module is added, the error increases linearly without sharp changes. This confirms that the SB-LSTM module helps to address the issues that arise during the training of internal flows with the existing model.

*4.4. Computation Time*

To evaluate the impact of the SB-LSTM module on the prediction time in deep learning, two comparisons were conducted. First, the time required for a deep learning model with the SB-LSTM module attached was compared with that of the deep learning model without the module. Second, the time required for the SB-LSTM model was compared with that of the conventional CFD calculation method. The calculation time was measured using the RTX 3090 GPU for the deep learning models, whereas a single core of the AMD Ryzen 5 5600G CPU was used for CFD calculations. In both tests, the average computation times were measured by performing the calculation on 1000 pieces of test data. In this study, the time required to solve the SIMPLE algorithm solver using the CFD method was measured utilizing a time command in Linux. The average calculation time was 1.16811 seconds. In the case of surrogate models, the experiment was set up to examine the impact of different batch sizes. A total of 12 experiments were conducted, with batch sizes of 1, 10, and 100. The results of each experiment are summarized in Table 3.

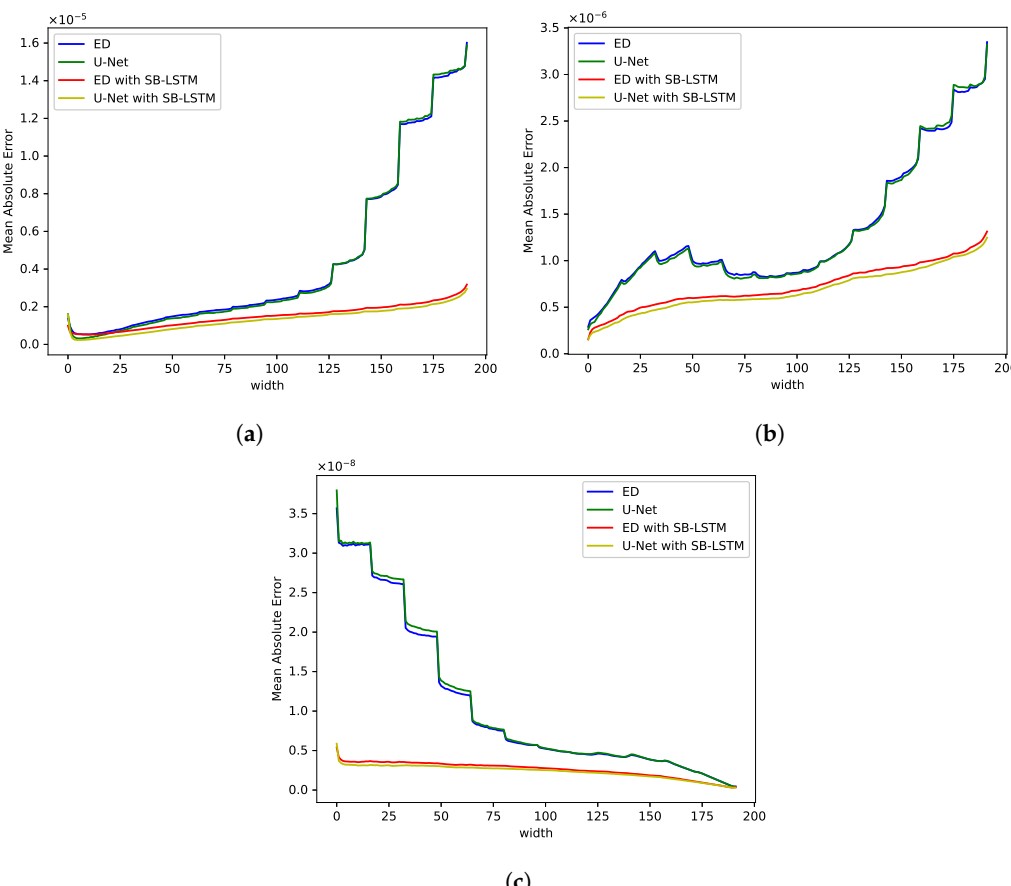

**Figure 15.** Figure of mean absolute error for each component of the models used in the experiment. From top left to bottom right: (**a**) $u$, (**b**) $v$, (**c**) $P$.

**Table 3.** Comparison of average computational time of four different methods in GPU environment.

| Method | Batch Size | Average Computation Time (s) | Inference Speed Acceleration Ratio |
|---|---|---|---|
| ED | 1 | $3.4044 \times 10^{-3}$ | 493.80 |
| | 10 | $2.0150 \times 10^{-3}$ | 834.3 |
| | 100 | $1.8760 \times 10^{-3}$ | 896.11 |
| U-Net | 1 | $4.4070 \times 10^{-3}$ | 381.46 |
| | 10 | $2.6121 \times 10^{-3}$ | 643.58 |
| | 100 | $2.4786 \times 10^{-3}$ | 678.25 |
| ED with SB-LSTM | 1 | $4.2567 \times 10^{-3}$ | 394.93 |
| | 10 | $2.2202 \times 10^{-3}$ | 757.1 |
| | 100 | $1.9821 \times 10^{-3}$ | 876.39 |
| U-Net with SB-LSTM | 1 | $4.9103 \times 10^{-3}$ | 342.36 |
| | 10 | $2.8264 \times 10^{-3}$ | 594.78 |
| | 100 | $2.5271 \times 10^{-3}$ | 665.23 |

For a batch size of one, the calculation time for the model with the SB-LSTM module was $0.8 \times 10^{-3}$ seconds longer than that for the ED model and $0.5 \times 10^{-3}$ s longer than that for the U-Net model, compared to the model without the SB-LSTM module. Although this is small considering that the CPU time required was 1.611 s, the SB-LSTM module can be considered inefficient in terms of calculation time as it took an additional 23% and 11% of the time compared to the baseline models. However, the additional time was due to the sequential calculation of the LSTM within the SB-LSTM module, which accounts for a large

portion of the total time when the batch size is one. As the batch size increases, the speedup value becomes comparable to that of the existing model.

## 5. Conclusions

This study was designed to determine whether the encoder–decoder or the U-Net structure, used for flow prediction, provides fast and acceptable results in learning internal flows compared to the existing CFD under various conditions. To this end, instead of generating arbitrary shapes between unchanged parallel walls, as in the existing method, we experimented with shapes created by simply joining serial pipes and CFD data corresponding to these shapes.

The existing neural networks exhibit problems in predicting flows from various patterns of shapes with different flow rates generated through the above method. To solve this problem, we proposed adding a module to the existing bottleneck part, to increase the learning capability through a spatial recurrent neural network that goes beyond the receptive field of the existing structure to capture information in the surrounding area, thereby solving the problems of the existing network, such as discontinuity in predicted values. When calculating the mean relative error in the region of interest, there was a decrease of at least 2.7% in the x-component of velocity, 4.7% in the y-component of velocity, and 15% in pressure, both in the overall area and the vortex corner region. Similarly, when comparing the computational time with the CFD method and the baseline model, this method was effective in terms of computational time.

During this research, the following shortcomings were discovered. In addition to evaluating the x-velocity and y-velocity used in the experiment as independent indicators, monitoring was also conducted on the direction created by both features. However, the directional plot did not align smoothly along the streamline; rather, it slightly jumped out like noise compared to the actual data flow. As a result, it seems necessary to conduct further research to force the interdependence between velocity fields, in addition to independently adjusting each data value.

Regarding follow-up studies, the dataset used in this work had a constant to fixed boundary condition. Therefore, follow-up flow prediction studies can add boundary condition information to the initial state of LSTM to study the effect of various boundary conditions. Additionally, flow prediction studies can be conducted for pipe shapes with a non-fixed domain size, taking advantage of the strengths of LSTM in language modeling.

**Author Contributions:** Conceptualization, J.K. and W.C.; methodology, J.K. and W.C.; software, J.K. and W.C.; validation, J.K. and W.C.; formal analysis, J.K. and W.C.; investigation, J.K. and W.C.; resources, J.K. and W.C.; data curation, J.K. and W.C.; writing—original draft preparation, J.K. and W.C.; writing—review and editing, J.K., W.C. and S.L.; visualization, J.K. and W.C.; supervision, S.L.; project administration, S.L. All authors have read and agreed to the published version of the manuscript.

**Funding:** This research received no external funding.

**Institutional Review Board Statement:** Not applicable.

**Informed Consent Statement:** Not applicable.

**Data Availability Statement:** Publicly available datasets were analyzed in this study. These data can be found here: [https://drive.google.com/file/d/1FY-TQfmuadkNOFuSsg1CbuB-kJ2I8612/view]. accessed on 19 October 2023.

**Conflicts of Interest:** The authors declare no conflict of interest.

## Appendix A. Qualitative Evaluation of SB-LSTM Attached Model

This section demonstrates that the SB-LSTM module does not interfere with the characteristics of existing models. To visualize the differences in the results of models with SB-LSTM attached, the same procedure as in Section 4.2 was used to plot the two models,

with the ground truth as a reference. The absolute error is also plotted below to visually confirm the points where there are differences.

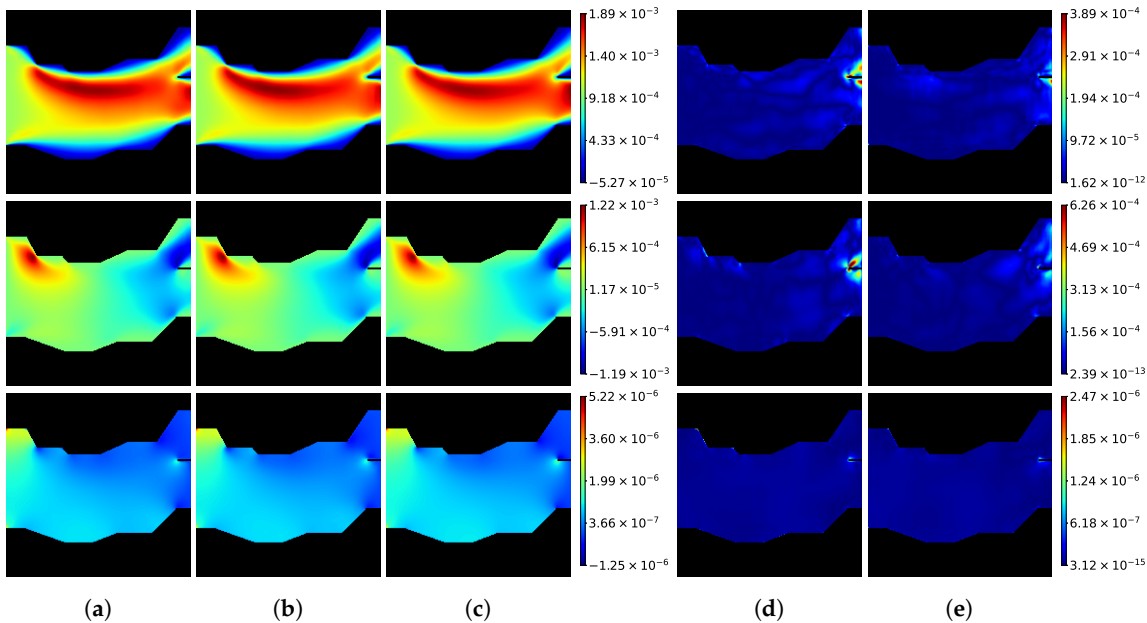

**Figure A1.** Qualitative results of four different methods. From top to bottom: $u$, $v$, $P$. From left to right: (**a**) ED with SB-LSTM, (**b**) U-Net with SB-LSTM, (**c**) ground truth, (**d**) absolute error of ED with SB-LSTM, (**e**) absolute error of U-Net with SB-LSTM.

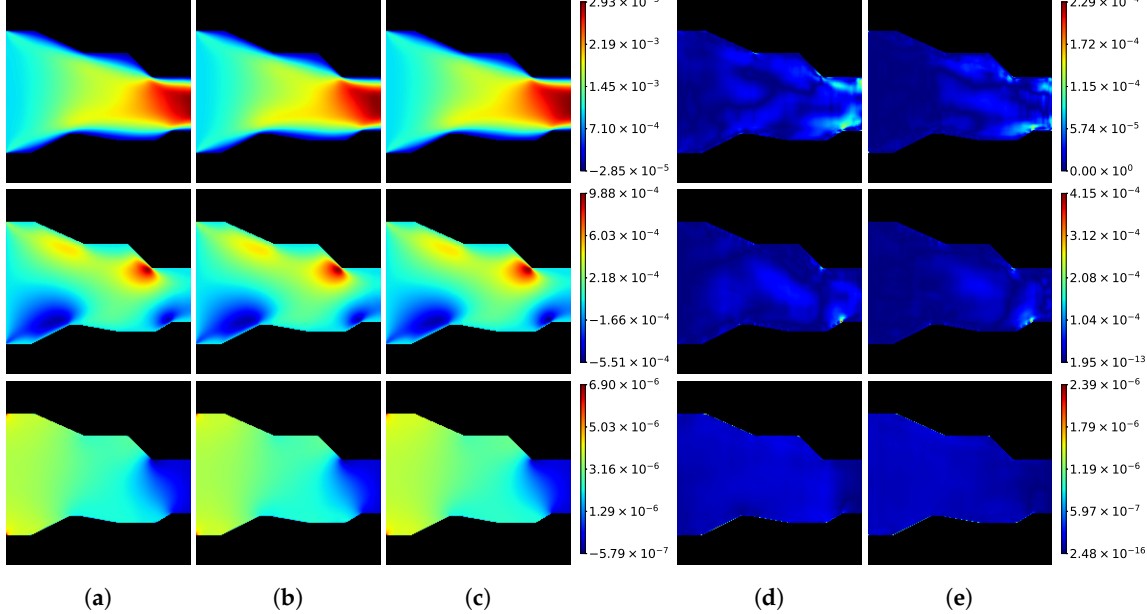

**Figure A2.** Qualitative results of four different methods. From top to bottom: $u$, $v$, $P$. From left to right: (**a**) ED with SB-LSTM, (**b**) U-Net with SB-LSTM, (**c**) ground truth, (**d**) absolute error of ED with SB-LSTM, (**e**) absolute error of U-Net with SB-LSTM.

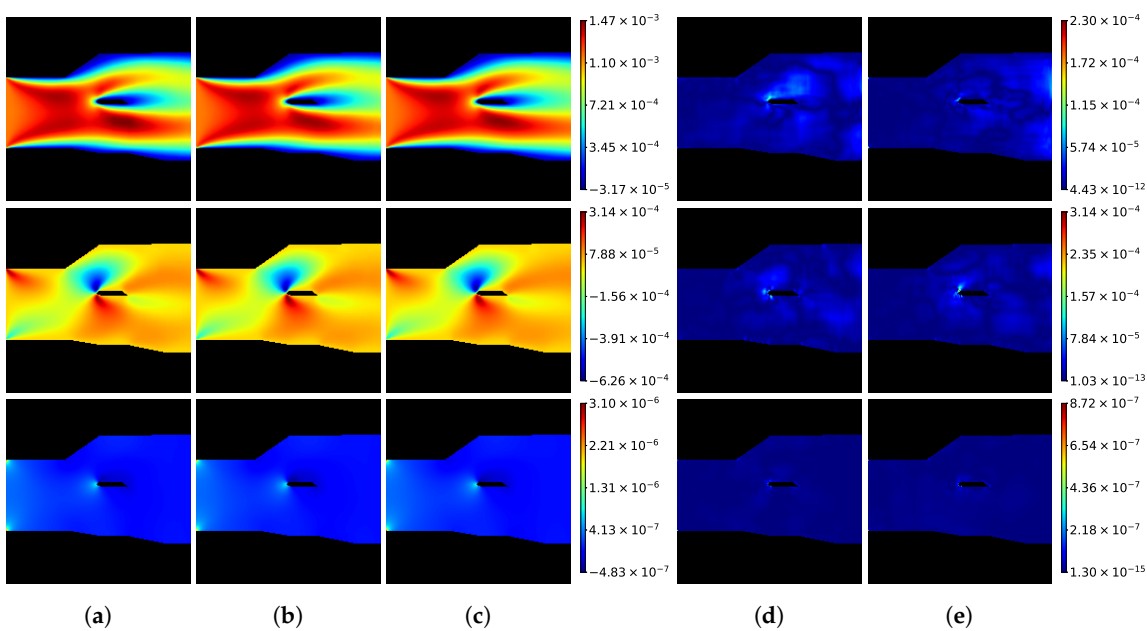

**Figure A3.** Qualitative results of four different methods. From top to bottom: $u$, $v$, $P$. From left to right: (**a**) ED with SB-LSTM, (**b**) U-Net with SB-LSTM, (**c**) ground truth, (**d**) absolute error of ED with SB-LSTM, (**e**) absolute error of U-Net with SB-LSTM.

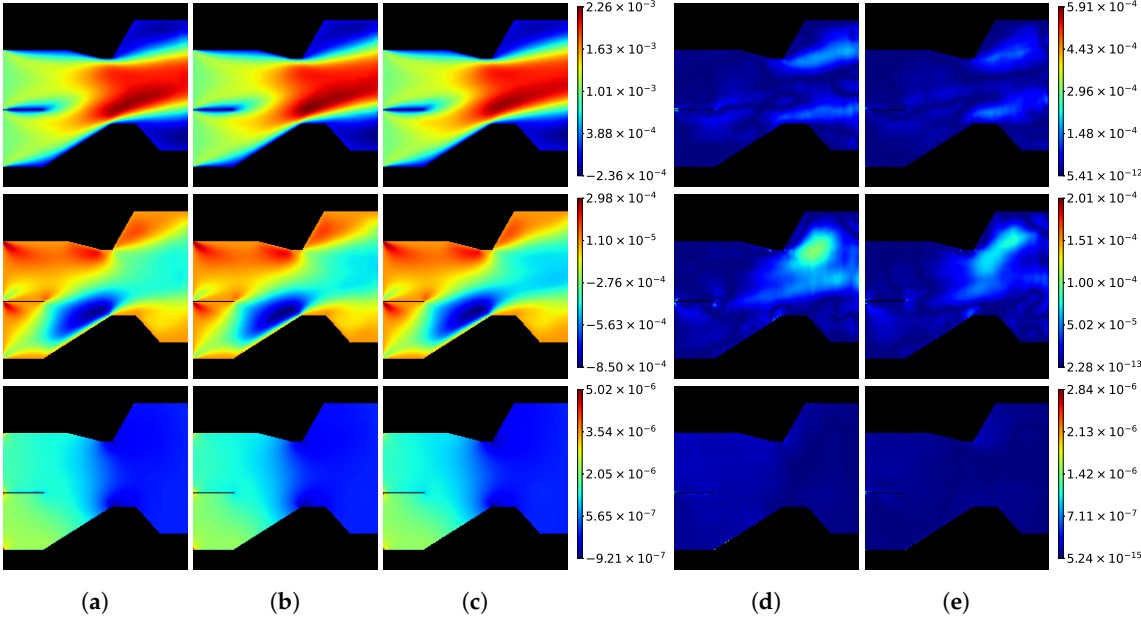

**Figure A4.** Qualitative results of four different methods. From top to bottom: $u$, $v$, $P$. From left to right: (**a**) ED with SB-LSTM, (**b**) U-Net with SB-LSTM, (**c**) ground truth, (**d**) absolute error of ED with SB-LSTM, (**e**) absolute error of U-Net with SB-LSTM.

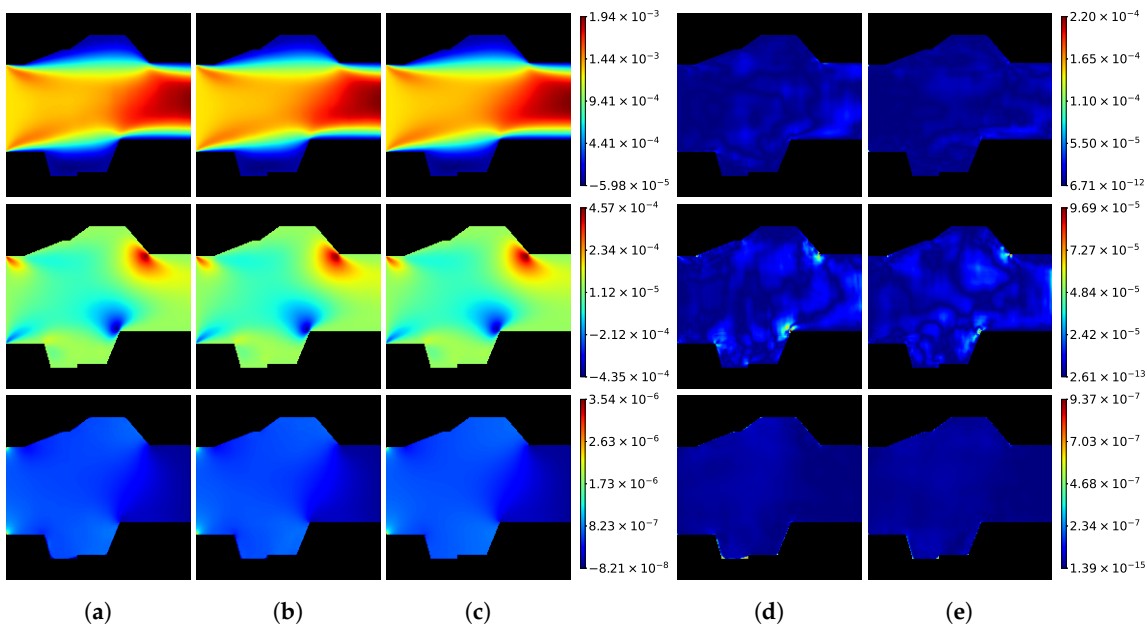

(a)       (b)       (c)       (d)       (e)

**Figure A5.** Qualitative results of four different methods. From top to bottom: *u*, *v*, *P*. From left to right: (**a**) ED with SB-LSTM, (**b**) U-Net with SB-LSTM, (**c**) ground truth, (**d**) absolute error of ED with SB-LSTM, (**e**) absolute error of U-Net with SB-LSTM.

We comprehensively evaluated the error plots of the cases visualized above, and we observed that the overall error value was lower in models with skip connections, and the errors occurring at the channel boundaries also decreased. This is the same as the difference shown in learning between the U-Net structure and the ED structure in [27].

## Appendix B. Comparison of Learning According to Lateral Connection Direction

The results of an additional experiment are included to examine the impact of directionality in the latent space on learning. In this experiment, we replaced the LSTM in the SB-LSTM module with a unidirectional LSTM to investigate the difference in learning. The only difference between the two experiments was the sequential process direction in the latent space, using either forward LSTM or backward LSTM for the experimental condition. The number of parameters in both experiments was the same. The training was conducted using the same hyperparameters as in the previous experiment.

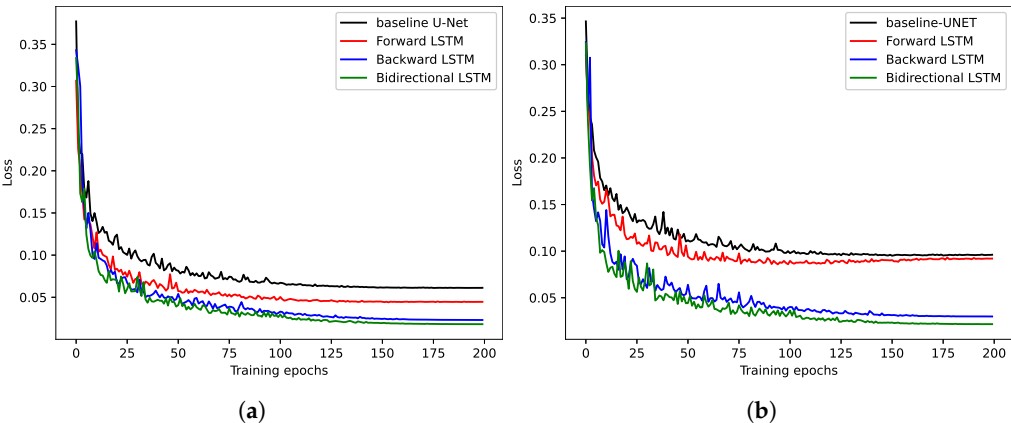

(a)                              (b)

**Figure A6.** Comparison of validation loss from LSTM with different directions. From left to right: (**a**) average loss, (**b**) pressure loss.

When LSTM was added, there was a common effect that sudden discontinuous intervals of values, as shown in Figures 10 and 14, disappeared in the case of velocity

prediction. However, in the case of pressure, the effect showed a significant difference depending on the direction, as shown in Figure A6. Upon analyzing the validation curve of the pressure in the training history, it was observed that even though the same number of learnable parameters was used, the forward LSTM overfitted after 90 epochs, whereas the learning of the backward LSTM was similar to that of the bidirectional LSTM. This indicates that simply adding LSTM does not guarantee improved performance. When incorporating a lateral connection, it is important to establish a pathway to receive the necessary information during supervised learning. An example result is as follows.

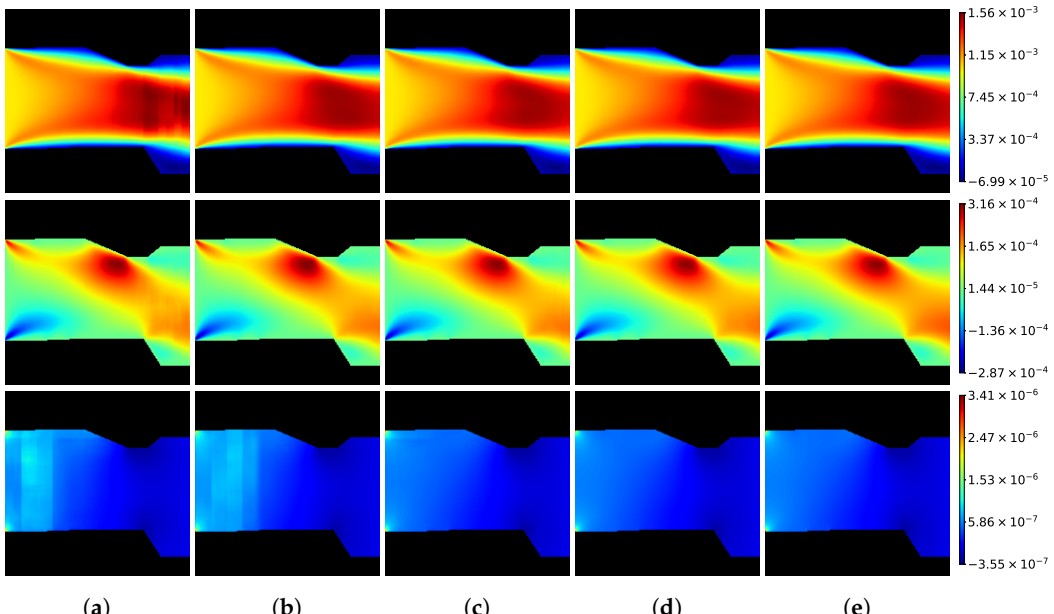

(**a**) (**b**) (**c**) (**d**) (**e**)

**Figure A7.** Qualitative results of four different methods. From top to bottom: *u*, *v*, *P*. From left to right: (**a**) U-Net, (**b**) U-Net with FLSTM, (**c**) U-Net with BLSTM, (**d**) U-Net with SB-LSTM, (**e**) target.

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
