# Peer review of "Internal Flow Prediction in Arbitrary Shaped Channel Using Stream-Wise Bidirectional LSTM"

_applsci, doi:10.3390/app132011481_

Round 1

Reviewer 1 Report

Comments to the manuscript entitled ‘Internal Flow Prediction on Arbitrary Shaped Channel Using Stream-wise Bidirectional LSTM’

The manuscript introduced a use of deep learning (DL) methods in CFD fields. The biggest advantage is its significantly shorter time. To broaden the application of DL in CFD, a new approach has been introduced, which involves a stream-wise bidirectional (SB)-LSTM module. This module is designed to generate a better latent space from the internal fluid region and additionally extract lateral connection features. this new DL approach in CFD, utilizing the SB-LSTM module, has shown promise in improving the accuracy and efficiency of fluid flow predictions, especially for internal flows like those found in pipes or vehicle engines.

The manuscript showed some valuable numerical results. However, there are still much to be improved before this manuscript can be published. Here are some comments.

1. Fig.1 and 2. How these geometry are generated and how can they be representatives of the Arbitrary Shaped Channel?

2. Algorithm Table 1. There is no need to use the table show the algorithm, it is better to use a diagram to show how this works.

3.In the simulation, the gravity is neglected. Could you explain why gravity is not considered ?

4.What about the mesh generation and turbulent models used in the study?

5. In total, 4 different methods are used. Why (a) ED, (b) U-Net, (c)

ED with SB-LSTM, (d) U-Net with SB-LSTM these four methods are used, is there any reason for the selection of the four methods?

6. Line 284, table 7 should be table 2.

7. Now you are doing simulations on 2D, how can this be applied to a more complex structure like 3D or with real geometry?

Minor editing of English language required

Reviewer 2 Report

This paper combines fluid mechanics and deep learning methods, and introduces a stream-wise bidirectional (SB)-LSTM module to improve the accuracy of the model. The effectiveness of the proposed method was verified through experiments. This paper has a reasonable framework and rich content, which requires minor modifications before publication. Some opinions are as follows:

1. The abstract section needs to be modified, and the conclusion section in the abstract is currently too broad. Suggest adding some quantitative statements in the conclusion section to demonstrate the effectiveness of the method.

2. What is the motivation behind the author's introduction of the stream-wise bidirectional (SB)-LSTM module? In other words, why did you think of using this module?

3. There is relatively little introduction to the LSTM model in the introduction section, and it is recommended to supplement it. For example:

A multi direct and indirect strategy for predicting wind direction based on the EMD-LSTM model.

4. The format needs to be modified, for example, -3 in line 326 needs to be superscripted.

5. The conclusion suggests adding the shortcomings of this method and future research directions.

Minor editing of English language required.

Reviewer 3 Report

The current paper tries to predict flow through a 2D pipe by using image processing methods, namely a U-net with an attached Bi directional - LSTM. This method performs better than standard encoder decoders. The results seem to back the authors claim and the study is detailed in nature, but the innovativeness of the study might be incremental, given that the study aims to predict real-time simulation data, but uses a huge amount of CFD data to do the same. 

Line 3 : his sentence doesn't make sense. Do you mean to say the lack of symmetry in the flow can cause difficulties in the prediction capabilities ?

Line 11 : please correct this sentence. Doesnt make sense

line 12: Claims are missing key numbers. An error metric/improvement metric is needed

Line 26: True, but none of them are stand-alone simulation techniques. They need pre existing CFD data to train. Which doesn't exclude the need of running sims.

Line 80: Not completely true. Please look at Turbulence flow based PINN type papers here. eg: "RANS-PINN based Simulation Surrogates for Predicting Turbulent Flows" and cited references

For the validation error, the LSTM+Unet only seems to do better for the pressure loss. Can the reason be specified ? Also, a pixel to pixel error map would help here.

The LSTM part does seem to improve performance of baseline ED/U-net type models, but more understanding is needed about the long short term memory angle. Mostly time varying cases are seen to improve because of these layers. How will an RNN approach do here ? Can any relevant literature help us understand this ? 

The NS equations contain a time varying unsteady differential, does the solution contain time variation, or is it steady state ? was a time dependent version of SIMPLE used ? 

Quality of language seems fine. 

Reviewer 4 Report

Your work and findings hold great promise in advancing the field of fluid dynamics and deep learning applications.

After a thorough review of the manuscript, the following comments should be addressed to strengthen the manuscript before it is considered for publication:

  1. The authors could provide more specific information about the dataset used in the experiments. What are the characteristics of this dataset, and how does it relate to the challenges of internal flow analysis? 
  2. The authors might emphasize the research gap more explicitly by discussing the limitations of previous approaches in handling internal flow analysis. 
  3. The authors could provide a deeper rationale for introducing the stream-wise bidirectional (SB)-LSTM module. The authors should explain why this specific approach was chosen and how it addresses the identified research gap. 
  4. The author could clarify how the SB-LSTM module was integrated into the existing models and how it improved their performance. 
  5. In the section on Dataset and Pre-Processing, it is beneficial to provide more detail about the specific characteristics of the shapes, especially if there are variations or unique features among them.
  6. The section on Model Parameters effectively communicates the increase in model capacity due to the addition of the SB-LSTM module. It would be valuable to briefly explain why the number of parameters increased, especially the role of fully connected (FC) layers in the bidirectional LSTM.
  7. While the section of Training Stability mentions that both the baseline models and models with the SB-LSTM module exhibited good training stability without overfitting, it would be beneficial to provide additional details on the metrics or techniques used to assess training stability. Did the authors monitor any other indicators of overfitting, such as validation accuracy or convergence speed?
  8. The visualizations in Figure 7 provide an excellent overview of training history. However, a brief discussion of any observed differences or nuances in the learning curves between the models would enhance the interpretation. Were there any specific phases during training (e.g., rapid convergence, plateaus) worth mentioning?
  9. The authors could provide a more detailed qualitative assessment of the visualized predictions. What specific differences or improvements can be observed when comparing models with and without the SB-LSTM module? Are there any notable features or artifacts in the predictions that should be discussed?
  10. Regarding the Comparison with SB-LSTM Models section, Since the focus is mainly on comparing models with and without the SB-LSTM module in this section, the authors should provide a clear and concise comparison of the qualitative results between these two groups. What are the key takeaways from this comparison, and how do they relate to the research objectives?

Overall, the manuscript is well-written, but these minor improvements, such as grammar and punctuation, clarity, and consistency, can enhance the quality of the English language.

Round 2

Reviewer 1 Report

The manuscript seems to be ready for publication.

Reviewer 4 Report

The authors have diligently addressed the referees' comments and suggestions properly.